# The central role of arginine in *Haemophilus influenzae* survival in a polymicrobial environment with *Streptococcus pneumoniae* and *Moraxella catarrhalis*

**Alexandra Tikhomirova**[1], **Peter S. Zilm**[2], **Claudia Trappetti**[1], **James C. Paton**[1], **Stephen P. Kidd**[1,3]*

**1** Department of Molecular and Biomedical Science, Research Centre for Infectious Diseases, University of Adelaide, Adelaide, Australia, **2** Department of Oral Microbiology, School of Dentistry, University of Adelaide, North Terrace Campus, Adelaide, South Australia, Australia, **3** Australian Centre for Antimicrobial Resistance Ecology, The University of Adelaide, Adelaide, Australia

* stephen.kidd@adelaide.edu.au

**Data Availability Statement:** All relevant data are within the paper and Supporting information files.

## Abstract

*Haemophilus influenzae*, *Streptococcus pneumoniae* and *Moraxella catarrhalis* are bacterial species which frequently co-colonise the nasopharynx, but can also transit to the middle ear to cause otitis media. Chronic otitis media is often associated with a polymicrobial infection by these bacteria. However, despite being present in polymicrobial infections, the molecular interactions between these bacterial species remain poorly understood. We have previously reported competitive interactions driven by pH and growth phase between *H. influenzae* and *S. pneumoniae*. In this study, we have revealed competitive interactions between the three otopathogens, which resulted in reduction of *H. influenzae* viability in co-culture with *S. pneumoniae* and in triple-species culture. Transcriptomic analysis by mRNA sequencing identified a central role of arginine in mediating these interactions. Arginine supplementation was able to increase *H. influenzae* survival in a dual-species environment with *S. pneumoniae*, and in a triple-species environment. Arginine was used by *H. influenzae* for ATP production, and levels of ATP generated in dual- and triple-species co-culture at early stages of growth were significantly higher than the combined ATP levels of single-species cultures. These results indicate a central role for arginine-mediated ATP production by *H. influenzae* in the polymicrobial community.

## Introduction

Otitis media (OM) is one of the most prevalent paediatric diseases, affecting 80% of children before the age of 3 [1]. OM frequently manifests as a chronic or recurrent infection [2], where conventional treatments such as antibiotic therapies [3], and tympanostomy tube placement [4] are often ineffective. The most common aetiologic agents of OM are the bacterial species *Streptococcus pneumoniae*, *Haemophilus influenzae* and *Moraxella catarrhalis* [5]. It is

**Funding:** This work was supported by the Garnett Passe and Rodney Williams Memorial Foundation Research Training Fellowship to A.T., and a National Health and Medical Research Council (NHMRC) Investigator Grant 1174876 to J.C.P. The funders had no role in study design, data collection and analysis, decision to publish, or preparation of the manuscript.

**Competing interests:** The authors have declared that no competing interests exist.

common that these species are present in the middle ear as a polymicrobial infection or in a polymicrobial biofilm [5,6]. They also regularly co-colonise the nasopharynx asymptomatically [7]. Concurrent carriage of all three pathogens has also been associated with clinical pneumonia [8].

We have previously identified the potential for both synergistic and competitive interactions between *S. pneumoniae* and *H. influenzae* in a dual-species polymicrobial environment, with the outcome of the interactions governed by environmental conditions, and characterised by a specific transcriptomic response [9,10]. However, the interactions between all three species, and the molecular aspects governing these interactions, have not been well defined [11], with current data showing both positive and negative interaction outcomes.

*M. catarrhalis* colonisation in children has been positively correlated to colonisation with *H. influenzae* [12], and another study has displayed positive associations of *S. pneumoniae* colonisation with co-colonisation by *M. catarrhalis* or *H. influenzae* [13]. However, a negative association between *H. influenzae* and *S. pneumoniae* has been reported during clinical studies of upper respiratory tract infection [14,15]. In addition, clinical studies have shown strain specificity to affect inter-species interactions [16], as well as significant variation in inter-species associations and species abundance between anatomical sites [5]. In a mouse colonisation model, co-infection of *S. pneumoniae* with *M. catarrhalis* was found to influence OM more significantly in comparison to co-infection of *S. pneumoniae* with *H. influenzae* [17]. In the context of antibiotic resistance of OM, *M. catarrhalis* has been shown to protect amoxicillin-susceptible *H. influenzae* and *S. pneumoniae* [18]. However, *S. pneumoniae* has also been shown to inhibit *M. catarrhalis* growth in a manner dependent on $H_2O_2$ and bacteriocin production [19].

The current study aims to elucidate the interactions between *H. influenzae*, *S. pneumoniae* and *M. catarrhalis*, and to assess the transcriptomic response of *H. influenzae* to growth in a triple-species co-culture with *S. pneumoniae* and *M. catarrhalis*.

## Materials and methods

### Bacterial growth and storage

All bacterial strains were stored at -80˚C in 30% glycerol. For all assays the *H. influenzae* isolate 86-028NP from the nasopharynx of a pediatric patient with OM, or laboratory strain Rd KW20, *S. pneumoniae* serotype 3 OM middle ear isolate strain 11, and *M. catarrhalis* reference strain QC (23240) obtained from SA Pathology laboratory (originating from a lung infection) were used for analyses. *H. influenzae* was grown on heart infusion (HI) agar plates supplemented with 10% levinthals, *S. pneumoniae* was pre-cultured on blood agar plates and *M. catarrhalis* was pre-cultured on HI agar supplemented with 5% defibrinated horse blood [9]. For pre-culture, all three bacterial species were grown until mid-log phase. *H. influenzae* was cultured in 10 mL of HI medium supplemented with 10 μg/mL hemin and 2 μg/mL β-NAD, with shaking at 100 rpm. *M. catarrhalis* was pre-cultured in 5 mL HI media supplemented with hemin and β-NAD, as above. *S. pneumoniae* was pre-cultured HI media supplemented with hemin and β-NAD statically [10].

### Growth analysis

HI media supplemented with hemin and β-NAD and, where specified, adjusted to pH 8, or unadjusted (pH 7) was used for growth analyses. Cultures were grown until mid-log phase for each species ($OD_{600}$ = 0.2–0.25 for *H. influenzae*, 0.25 for *S. pneumoniae* and 0.2 for *M. catarrhalis*). Mono-culture, dual-culture and triple-culture inoculums contained 1/20 dilution of each species [20]. Cultures were grown statically in a 96-well microtitre plate. After 18 h of

growth, mono- and co-cultures were collected, and plated onto HI agar or selective blood agar plates containing gentamicin (selecting for *S. pneumoniae*) at a concentration of 5 μg/mL, bacitracin (selecting for *H. influenzae*) at a concentration of 300 μg/mL, and HI agar supplemented with blood and vancomycin at a concentration of 3 μg/mL (selecting for *M. catarrhalis*). All growth assays were performed in triplicate.

## Growth analysis with arginine

To assess *H. influenzae* growth with arginine, *H. influenzae* was inoculated into 10 mL of HI media containing no supplemented arginine or supplemented with an additional 2 or 4 g/L L-arginine (Sigma). The pH of the media with and without arginine was adjusted to pH 8 or 7 as specified, and cultures were grown statically or with shaking at 100 rpm for 18 h. For growth analysis $OD_{600}$ was measured every 30 min for 18h on a SPECTRAmax spectrophotometer (Molecular Devices).

## ATP production assay

ATP production was measured using the BacTiter-Glo Microbial Cell Viability Assay (Promega, Australia). To measure *H. influenzae* ATP production in the absence and presence of arginine, *H. influenzae* 86-028NP was pre-cultured to mid-log phase. Subsequently, the pre-culture was inoculated in a 1/20 dilution into 5 mL of HI media with 0, 2, 4 or 6 g/L arginine. Every 15 min, 50 μL of the culture was added to 50 μL of the BacTiter Glo reagent in a white 96-well plate, and luminescence was measured on the Synergy HTX Spectrophotometer (Biotek). $OD_{600}$ of the culture was concurrently measured to ensure ATP production was not related to growth.

ATP production in single-species, dual-species and triple-species cultures was also measured using BacTiter-Glo Microbial Cell Viability Assay on the PHERAstar Spectrophotometer. Cultures were inoculated for 2h, following which ATP production was measured. Cultures were concurrently plated on selective media to ensure ATP production was unrelated to growth.

## Intracellular pH assay

Intracellular pH was determined using the pH sensitive dye BCECF-AM (Invitrogen) using a protocol established for measuring intracellular pH in bacteria [21]. Cells were resuspended in PBS containing 1 μM BCECF-AM, followed by 30 min incubation at 37˚C [21], washed in PBS to remove extracellular dye, and subsequently resuspended in 5 mL PBS. Fluorescence was measured using an excitation wavelength of 490nm and an emission wavelength of 530 nm on the (PHERAstar spectrophotometer, BMG LabTech). An intracellular pH standard curve was generated by using the intracellular pH calibration buffer kit (Invitrogen), according to the manufacturer's guidelines.

## RNA preparation

RNA sequencing was performed on cultures following 2 h of growth. Following pre-culture, 4 mL aliquots of mono-species, dual-species and triple-species cultures were prepared, in triplicate, as biological replicates of cell growth. Cultures were grown at 37˚C for 2 h.

Following this, cultures were transferred directly to an equal volume of RNA Protect Bacterial Reagent (Qiagen), vortexed, incubated for 5 min at room temperature, and centrifuged ($4000 \times g$ for 10 min) [9,10]. The resulting cell pellet was stored at −80˚C. For the RNA

extraction, a combination of the hot phenol extraction method and a commercial RNA extraction kit (Qiagen, RNeasy Mini Kit) were utilised, as we have previously performed [9,10].

## Transcriptomics analysis

RNA from each biological replicate was supplied to the Adelaide Cancer Genomic Research Facility (Adelaide, Australia) for library preparation and sequencing (RNAseq) using the Illumina NextSeq platform (Illumina). The fastq files were aligned to the reference genome of 86-028NP (for *H. influenzae* Genbank: NC_007146), using bowtie2 in the local mode. The resulting sam files were converted to bam file format, and then to the bed file format. Subsequently, the bed files were aligned to the reference genomes in gff format. The statistical analysis was performed with the DESeq package in R, to identify *H. influenzae* genes with the largest shift in expression in co-culture conditions compared to mono-culture conditions [9,10]. Differentially expressed genes presented in S1–S4 Tables display statistically significantly differentially expressed genes (p<0.01), which display a fold change in expression >2. The raw transcriptomic data has been made publicly available on the NCBI SRA database with accession number PRJNA820503 and submission ID SUB11224062.

## Gene expression confirmation by qRT-PCR

Gene expression of *artM* was confirmed by a one-step relative qRT-PCR in a Roche LC480 real-time cycler, as we have performed previously [22]. RNA extracted as previously described from bacterial single-species, dual-species and triple species cultures following 2 h of growth was used to confirm gene expression by qualitative real-time PCR.

Primers used to amplify *H. influenzae artM* (CAAGAATATTTAAGCGTGATCG; AGTAAGG TATAAATTGACCGCA) and 16SrRNA (TGGCAACAAAGGATAAGGGTT; TCCTAAGAAGAGC TCAGAGAT) were used at a final concentration of 200 nM. *H. influenzae artM* and 16S rRNA primers were tested for absence of amplification of *artM* and 16S rRNA of mono-species *S. pneumoniae* and *M. catarrhalis* cultures. Amplification data was analysed using the comparative critical threshold method, as previously described and is presented as a percentage of total expression relative to 86-028NP [22]. Assays were performed in triplicate in two experiments. Statistical analyses were performed using the t-test, *p<0.05.

## Results

### There is strain-specific competition with *H. influenzae* in a dual-species environment with *S. pneumoniae* and in a triple-species environment with *S. pneumoniae* and *M. catarrhalis*

Our work has previously assessed the co-existence of *H. influenzae* strain Rd KW20 and *S. pneumoniae* in a polymicrobial environment and has identified both competitive and synergistic interaction outcomes [9,10]. In this study we aimed to assess the interactions of *H. influenzae* in a triple-species environment harbouring *M. catarrhalis* as well as *S. pneumoniae*. We have previously identified the interactions between *H. influenzae* and *S. pneumoniae* to result in either a synergistic or competitive outcome, as a consequence of specific parameters, including pH, growth phase and nutrient availability [9,10]. Hence, four analyses in this study, we used a pH of 8 for the growth media, conditions which we have previously shown resulted in reduced species competition [9]. For our current research we assessed both the *H. influenzae* laboratory strain Rd KW20 as was used in our previous analysis, as well as the genetically distinct, nontypeable, clinical OM isolate 86-028NP.

Determination of CFUs following 18 h growth in mono- or co-culture, showed that both strains were negatively affected by co-culture with *S. pneumoniae* and by co-culture in a triple-species environment, both in planktonic and biofilm form (Fig 1). However, strain 86-028NP was more severely affected than Rd KW20, reaching cell numbers below the detection limit in planktonic co-culture with *S. pneumoniae* (Fig 1A), and significantly lower cell numbers

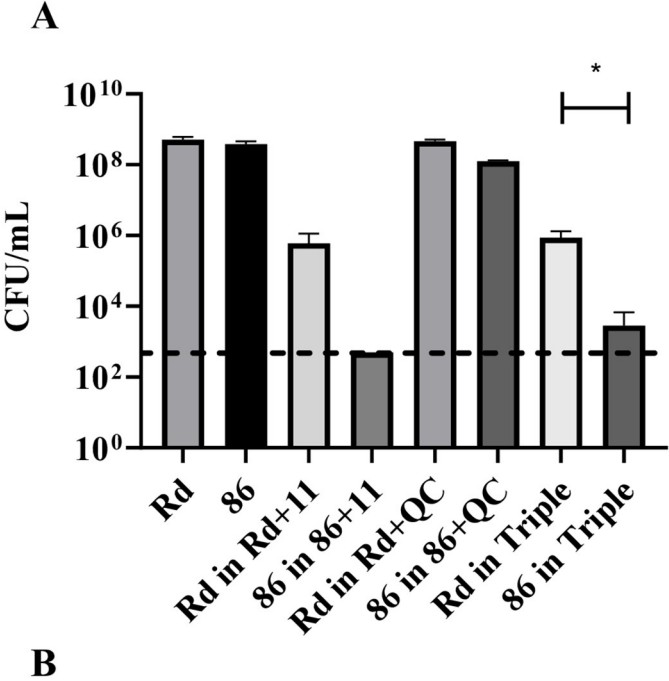

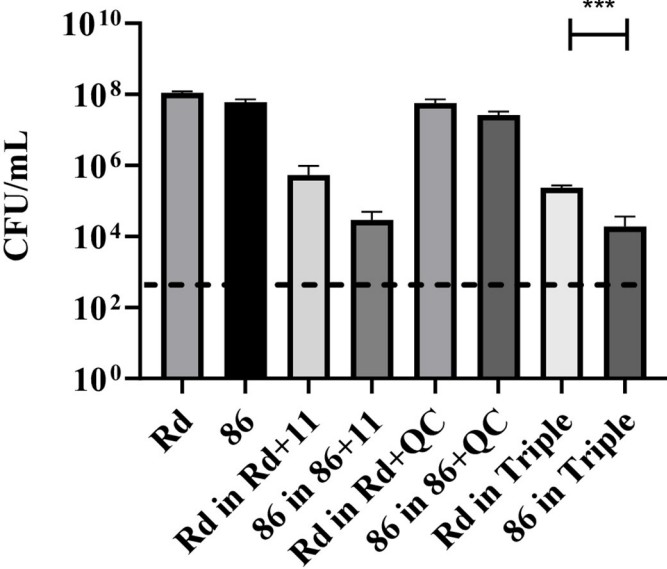

**Fig 1. Growth of *H. influenzae* in dual- and triple-species co-cultures with *S. pneumoniae* and *M. catarrhalis* after 18 h.** Growth in (A) planktonic and (B) biofilm state of *H. influenzae* middle ear isolate 86-028NP and laboratory strain Rd KW20 in mono-culture, in co-culture with *S. pneumoniae* strain 11, in co-culture with *M. catarrhalis* strain QC, and in triple-species co-culture with *S. pneumoniae* 11 and *M. catarrhalis* QC after 18h is represented as CFU/mL. * $p < 0.05$, *** $p < 0.001$.

compared to Rd KW20 in triple-species co-culture. While Rd KW20 was affected to a similar level by co-culture in planktonic and biofilm state, 86-028NP was more affected by co-culture in the planktonic state, whereas in a biofilm, it had a better capacity to co-exist in dual- and triple-species culture (Fig 1).

## Competitive interactions with *H. influenzae* in dual- and triple-species environments are time-point specific

As we have previously shown that there were time-dependent competitive interactions between *H. influenzae* Rd KW20 and *S. pneumoniae* [9], to assess whether the inhibitory effects of *S. pneumoniae* or a triple-species environment on strain 86-028NP were also time-dependent, growth of each species was assessed in planktonic culture at early log phase, following 2 h of growth (Fig 2). The results showed that all three species displayed an equivalent growth in mono-, dual- and triple-species co-culture (Fig 2).

## *H. influenzae* up-regulates arginine uptake in dual- species co-culture with *S. pneumoniae*, and in triple-species co-culture

Our previous analysis of the *H. influenzae* Rd KW20 transcriptome in co-culture with *S. pneumoniae* has identified molecular factors central to polymicrobial co-existence in distinct environmental conditions [9,10]. As 86-028NP was more affected by dual- and triple- polymicrobial co-culture than Rd KW20, in this study we endeavoured to characterise the strain-specific *H. influenzae* 86-028NP transcriptome, but within a triple-species context which is relevant to the OM scenario. For the transcriptomic assessment of molecular events facilitating *H. influenzae* survival in the dual- and triple-species conditions, we performed a transcriptomic analysis following 2 h of growth, a timepoint in which all the bacterial species were viable and prior to competition affecting inter-species interactions.

Transcriptomic analysis identified *H. influenzae* genes up- and down-regulated in dual- and triple-species co-culture compared to monoculture (S1–S4 Tables), and importantly, showed a consistent up-regulation of genes of the arginine uptake system *artM-artP* (*artM*, *artQ*, *artI* and *artP*) in *H. influenzae* in dual-species culture with *S. pneumoniae*, as well as in the triple-species co-culture (S1–S3 Tables). No genes were significantly differentially expressed in *H. influenzae* when in dual-species co-culture with *M. catarrhalis*.

The expression of the first gene of the *artM-artP* operon, *artM*, following *H. influenzae* growth for 2 h in dual-species and triple-species conditions was confirmed by qRT-PCR (S1 Fig), supporting *H. influenzae* having a requirement for arginine in a dual- and triple species environment.

## Arginine supplementation does not impact *H. influenzae* growth, but restores its survival in co-culture with *S. pneumoniae*, and in the triple-species culture

Assessment of arginine supplementation on growth phenotype of the mono-, dual- and triple-species cultures showed that the presence of exogenous supplemented arginine did not affect growth of *H. influenzae* or *M. catarrhalis*, but did produce a biphasic growth for *S. pneumoniae*, which was also reflected in the dual- and triple-species growth curves containing *S. pneumoniae* (S2 Fig).

To assess a necessity for arginine in conditions of polymicrobial culture, *H. influenzae* 86-028NP and Rd KW20 were separately grown in dual- and triple-species co-cultures in the absence and presence of exogenous arginine supplemented at 4 g/L (Fig 3), and viable cell

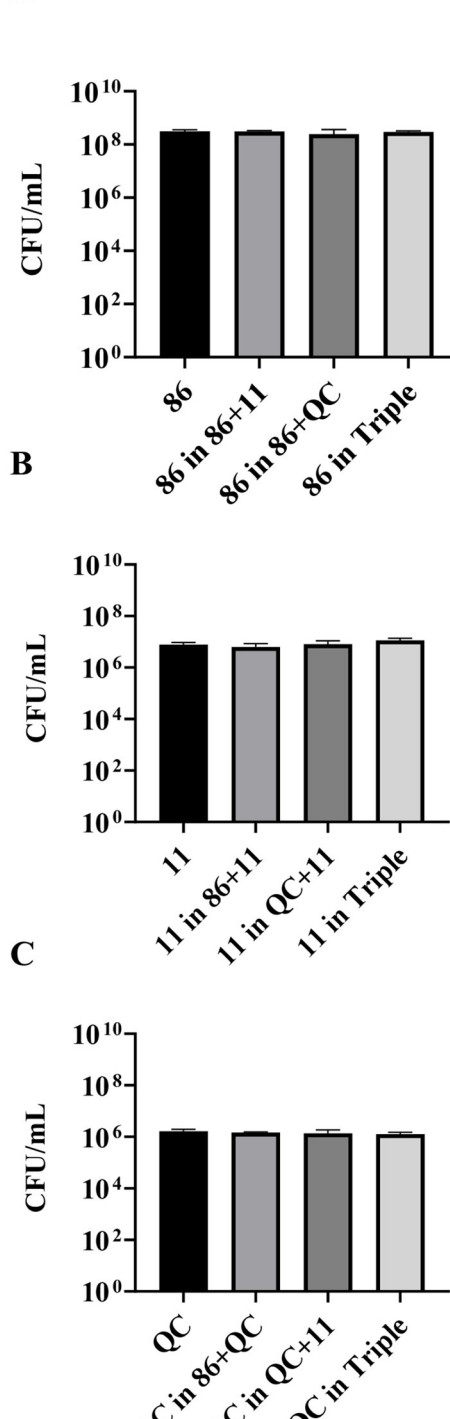

**Fig 2. Growth dynamics of dual- and triple-species co-cultures of *H. influenzae*, *S. pneumoniae* and *M. catarrhalis* after 2 h.** Growth in mono-, dual- and triple-species co-culture of (A), *H. influenzae* 86-028NP, (B), *S. pneumoniae* 11, and (C), *M. catarrhalis* QC in the planktonic state after 2h of growth are demonstrated as CFU/mL.

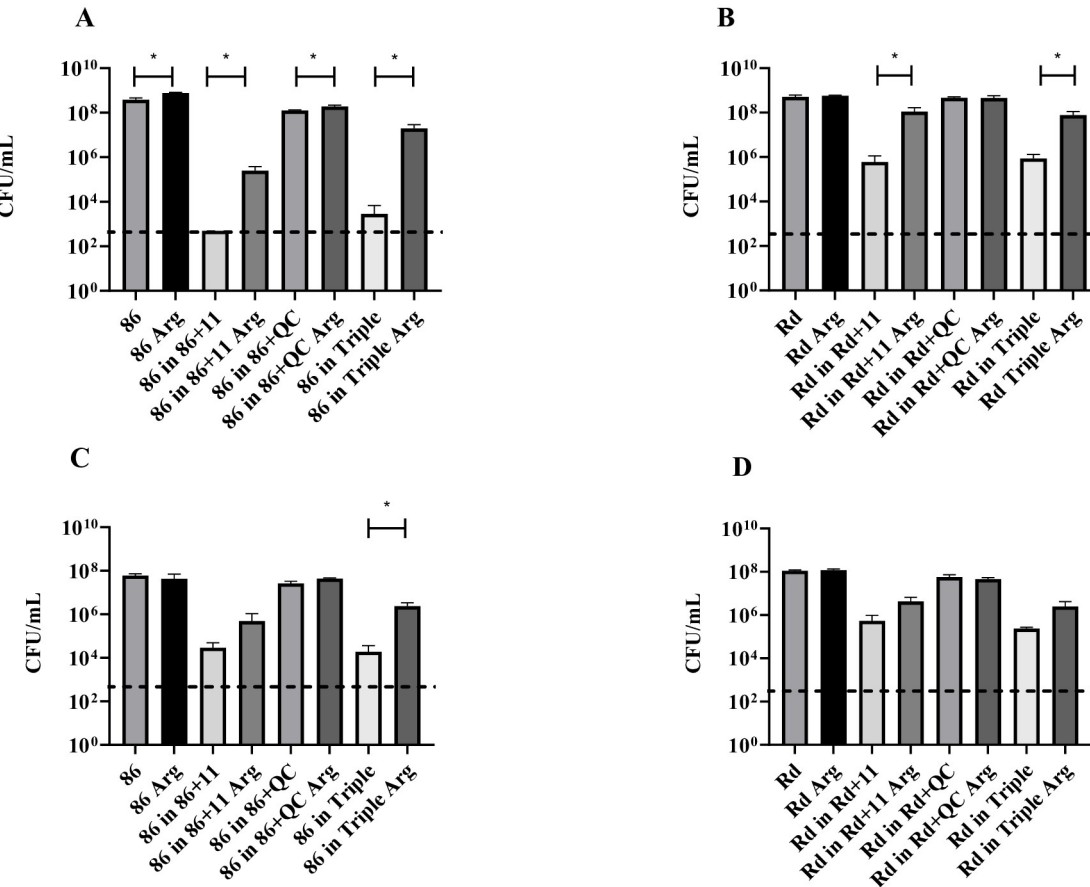

**Fig 3. Arginine restores *H. influenzae* survival in dual- and triple-species co-culture.** Viable cell numbers of *H. influenzae* Rd KW20 or *H. influenzae* 86-028NP in planktonic (A,B) and biofilm (C,D) growth in mono-culture, dual-culture with *S. pneumoniae* 11, dual-culture with *M. catarrhalis* QC, and in triple-species culture for 18h, with and without supplementation with exogenous arginine at a concentration of 4g/L. Dotted line indicates the limit of detection. *, $p < 0.05$.

counts were determined following 18 h growth. In both strains the presence of exogenous arginine enabled the recovery of *H. influenzae* survival in co-culture with *S. pneumoniae*, and in triple-species culture. However, the recovery effect of arginine was significantly more pronounced for strain 86-028NP than for Rd KW20 (Fig 3), and was greater in the planktonic lifestyle (Fig 3A and 3B) than in the biofilm lifestyle (Fig 3C and 3D).

## Arginine enhances *H. influenzae* ATP production

Uptake of arginine by *H. influenzae* could serve to enhance ATP production at a particular time-point during polymicrobial co-culture. We therefore measured ATP production (BacTiter Glo kit, Promega, Australia) at 15 min intervals with varying concentrations of exogenously supplemented arginine. Arginine significantly increased ATP production in 86-028NP during early lag phase of growth. Addition of higher arginine concentrations showed a trend for higher ATP production, although this was not statistically significant (Fig 4).

## ATP production is increased in dual- and triple-species cultures

As *H. influenzae* ATP production was related to the presence of arginine, we then assessed the ATP production in the dual- and triple-species polymicrobial culture after 2 h of growth,

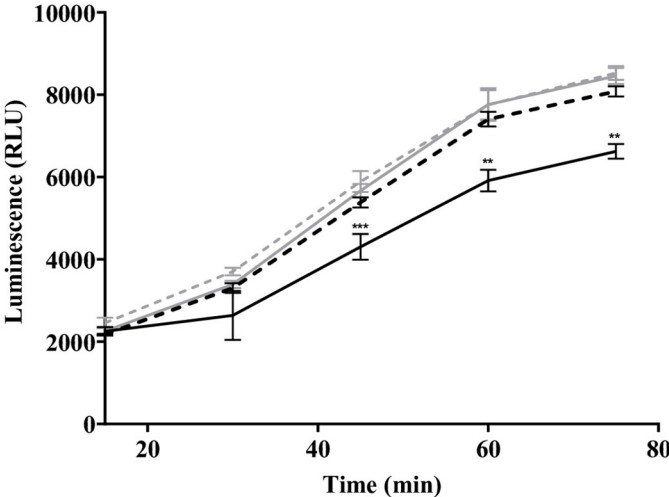

**Fig 4. Arginine supplementation enhances *H. influenzae* ATP production.** Relative ATP production measured by luminescence (relative light units, RLU), of *H. influenzae* without supplementation of arginine (black), and with supplementation of 2 g/L (black dotted line), 4 g/L arginine (grey filled line) and 6 g/L arginine (grey dotted line). Luminescence (ATP production) is significantly higher in all concentrations of arginine, compared to no arginine supplementation. Significant differences were determined with a student t-test (*** $p \leq 0.001$, ** $p \leq 0.01$).

correlating to the time-point arginine uptake was up-regulated, in comparison to the single-species cultures. Comparisons were performed in a growth medium at pH 8, as well as a growth medium at pH 7, conditions we have previously reported to result in higher levels of species competition [9]. ATP production was significantly higher for the *H. influenzae* and *S. pneumoniae* dual-species culture (86-028NP+11), and in the triple-species culture, compared to the *H. influenzae* single-species culture (86-028NP) (Fig 5). Importantly, the 86-028NP+11 dual-species culture, produced a significantly higher amount of ATP than the combined ATP for 86-028NP and 11 in single-species culture, and this was more pronounced at pH 7 (Fig 5B). The ATP production in triple-species culture was also significantly higher than the combined ATP production of the 86-028NP+11 in co-culture and QC in mono-culture or 86-028NP+QC dual-species culture and 11 in mono-culture (Fig 5). The presence of exogenously supplemented arginine did not significantly affect ATP production.

## Arginine does not impact *H. influenzae* intracellular pH

In other bacterial species, arginine has been shown to serve different cellular roles such as regulating the intracellular pH. Using the BCECF-AM intracellular pH dye, we performed an analysis of intracellular pH in *H. influenzae* following incubation with exogenously supplemented arginine. Supplementation with arginine did not result in a significant alteration of intracellular pH (S3 Fig).

## Discussion

In this study, we have utilized a model for the polymicrobial culture of *H. influenzae*, *S. pneumoniae* and *M. catarrhalis*, which has enabled the analysis of planktonic and biofilm cells within their mono-, dual- and triple-species communities. This expands on our previous publications on the co-culture of *H. influenzae* and *S. pneumoniae* [9,10].

For each species within dual- and triple-species conditions, growth was equivalent after 2 h of culture compared to mono-species culture. However, inter-species competition was

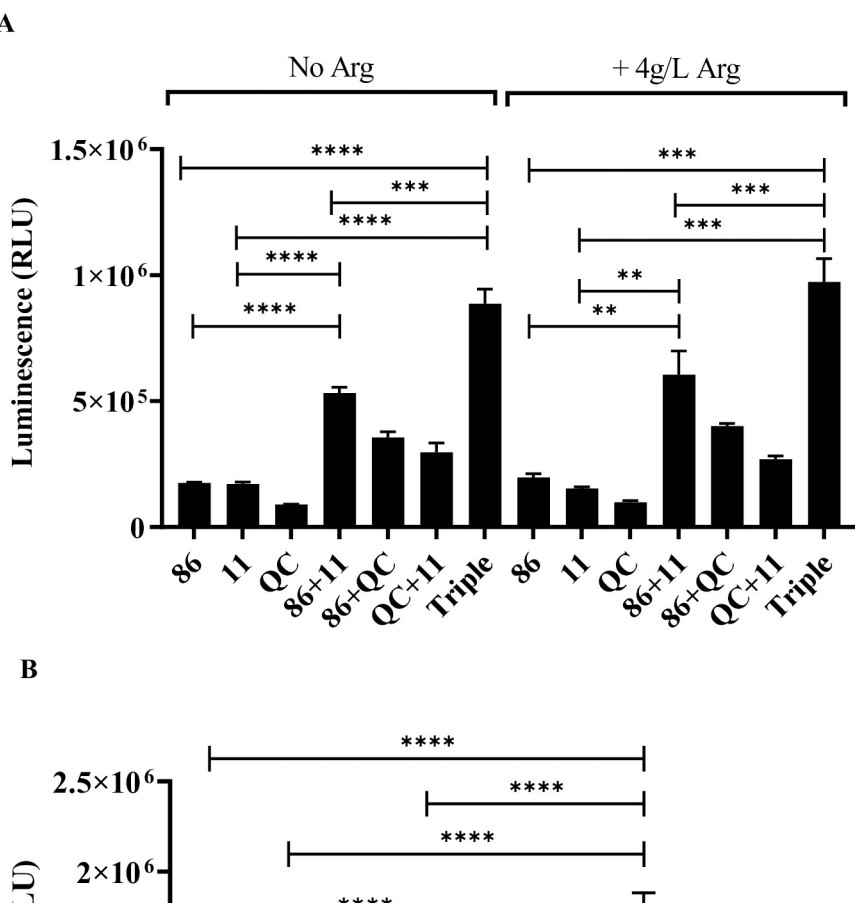

**Fig 5. ATP production is enhanced in dual- and triple-species cultures.** ATP production measured by luminescence (relative light units, RLU), of *H. influenzae* 86-028NP (86), *S. pneumoniae* 11 and *M. catarrhalis* QC following 2 h of incubation in single-species culture, dual-species culture, and triple species culture at A) pH 8, with and without arginine supplementation, and B) in pH 7, without arginine supplementation. Significant differences were observed between *H. influenzae* alone (86), in dual-species culture with *S. pneumoniae* (86+11), and in triple-species culture (Triple). Statistical significance was evaluated with a student's t-test (** $p < 0.01$, *** $p < 0.001$, **** $p \leq 0.0001$).

observed at 18 h of growth in both the planktonic and biofilm state, where *H. influenzae* was reduced in co-culture with *S. pneumoniae* and in the triple-species culture (Fig 1). This competitive effect was strain-specific; the *H. influenzae* clinical isolate 86-028NP was significantly more affected by dual-species culture with *S. pneumoniae*, and by triple-species culture, in

comparison to the laboratory strain Rd KW20. This result suggests a specific importance of competition for clinical *H. influenzae* isolates within the polymicrobial community.

We had previously established the *H. influenzae* Rd KW20 transcriptional response to co-culture with *S. pneumoniae*. In the current study we assessed the transcriptional response of *H. influenzae* 86-028NP in the presence of *S. pneumoniae* and *M. catarrhalis* in dual- and triple-species co-culture [9,10].

Genes of the arginine uptake operon were consistently up-regulated in *H. influenzae* both in dual-species co-culture with *S. pneumoniae*, and in the triple-species co-culture (S1 and S2 Tables). While *H. influenzae* is known to have a requirement for arginine and lacks the initial 5 steps of the arginine biosynthesis pathway, the role of arginine for *H. influenzae* survival in a polymicrobial environment has not been established [23]. Addition of exogenous arginine was able to partially reverse the competitive effect of co-culture on *H. influenzae*. Interestingly, arginine had a greater effect in recovering the survival of 86-028NP in co-culture compared to Rd KW20, and a more pronounced effect was observed in planktonic co-culture, compared to the biofilm state (Fig 3). These results highlight a critical role for arginine in enabling the co-existence of *H. influenzae*, *S. pneumoniae* and *M. catarrhalis*, and strain-specificity of this role.

These results imply that arginine may have a previously unrecognised role in nutrient sharing between the bacterial species. *M. catarrhalis* is unable to utilise extracellular carbohydrates [24], and is reliant on alternative carbon sources, including amino acids. While *M. catarrhalis* possesses genes for the synthesis of most amino acids, it does not possess pathways for the biosynthesis of arginine [24]. *S. pneumoniae* also lacks a complete pathway of arginine metabolism [25], and *H. influenzae* likewise lacks the initial 5 steps of the arginine biosynthesis pathway [23]. Therefore, it seems likely that competition driven by the ability to uptake arginine may result in this polymicrobial environment. Despite potential for such competition, growth phenotype analysis showed that only *S. pneumoniae* growth was different in the presence of exogenous arginine, resulting in a biphasic growth phenotype. This suggested arginine utilisation in *S. pneumoniae* following utilisation of a preferred nutrient source, but showed that arginine did not increase mono-species growth of *H. influenzae* or *M. catarrhalis* (S2 Fig).

Conceivably, arginine could be important for additional reasons, including ATP generation, or acidity regulation. In other bacterial species, arginine was found to enhance acid tolerance and/or generate ATP via the arginine deiminase pathway [26]. The arginine deiminase pathway comprises a three-step reaction, catalysed by arginine deiminase, ornithine transcarbamoylase and carbamate kinase, respectively [27] (Fig 6). The reaction converts arginine to ornithine, ammonia and carbon dioxide, generating 1 mol of ATP per 1 mol of arginine. In this context, ammonia acts to buffer the environmental pH (25). However, while *H. influenzae* possesses both the ornithine transcarbamoylase and carbamate kinase, it does not possess the arginine deiminase gene [27], therefore in the absence of an orthologous enzyme which could catalyse the reaction, *H. influenzae* would not be expected to produce ammonia or ATP through this pathway (Fig 6).

Despite the lack of a complete arginine deiminase pathway, the presence of supplemented arginine did result in enhanced ATP production in *H. influenzae* (Fig 4). In the context of nutrient limitation and competition in a polymicrobial environment, ATP production in *H. influenzae* may be critical to continue to survive in both a dual-species and triple-species environment. Indeed, ATP production was shown to increase in the dual-species and triple-species environment of *H. influenzae*, *S. pneumoniae* and *M. catarrhalis* (Fig 5). This indicates that the condition of polymicrobial co-culture stimulates ATP production in *H. influenzae*, and potentially also stimulates ATP production in *S. pneumoniae* and *M. catarrhalis*. Supplementation with exogenous arginine did not enhance ATP production, suggesting that the polymicrobial community and not arginine levels drives the ATP production in co-culture. The increase in

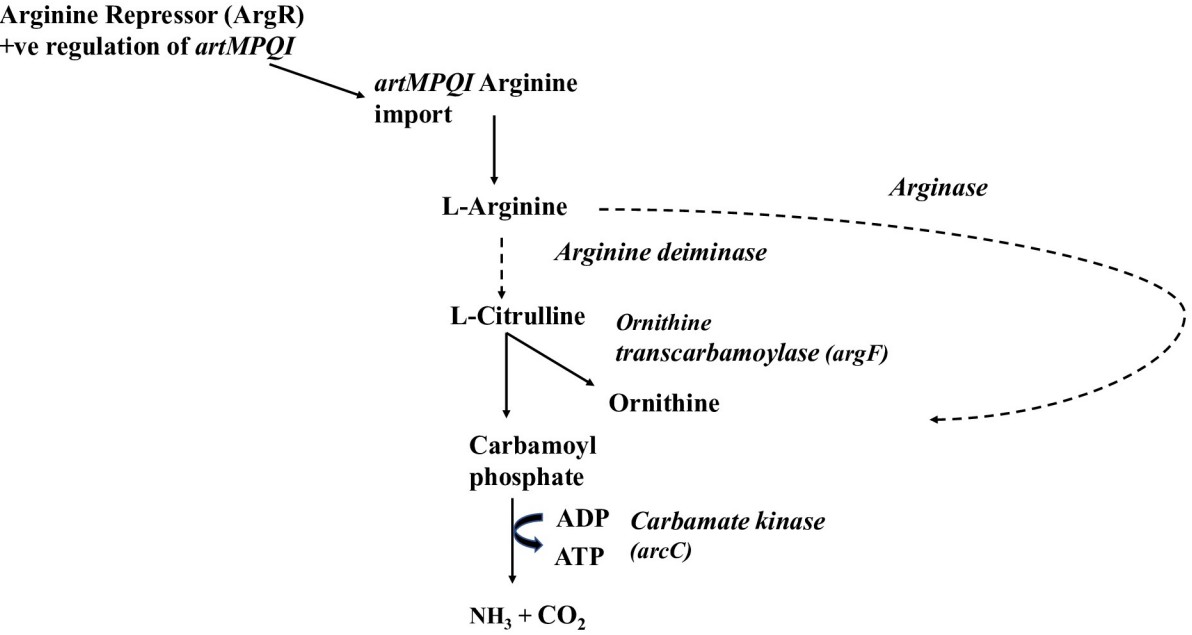

**Fig 6. Arginine import and catabolic pathways in *H. influenzae* 86-028NP as determined from KEGG pathway analysis.** Pathways for which genes have been identified in 86-028NP are indicated with black line, and pathways for which genes are missing in 86-028NP are indicated with dashed line.

ATP generation in *S. pneumoniae* and *H. influenzae* dual-species culture and triple-species culture was greater at pH 7, where competitive interactions have previously been observed to be greater, suggesting that early ATP generation may be necessary for *H. influenzae* viability in the polymicrobial environment [9].

In other bacterial species, ATP production from arginine utilisation is achieved primarily through the arginine deiminase pathway. The arginine deiminase pathway is also important in the pH regulation of bacteria, as ammonia is produced in the last step, resulting in alkalisation of the intracellular pH (25). As arginine utilisation was induced in *H. influenzae* in conditions of co-culture with *S. pneumoniae*, where we have previously shown acid stress to play a role [9], arginine, through a modified arginine deiminase pathway, could potentially result in alkalization of intracellular pH. However, our results have indicated that the presence of supplemented arginine did not affect the intracellular pH of *H. influenzae* (S3 Fig). Importantly, our transcriptomic data showed a concurrent increase in expression of an aspartate ammonia lyase in co-culture with *S. pneumoniae*, and in triple species co-culture (S3 Table), as well as *artM*. Aspartate ammonia lyase can neutralise the produced ammonia, by catalysing the conversion of ammonia and fumaric acid to aspartate [28], therefore utilisation of arginine through a similar pathway may not necessarily result in detectable increases in intracellular pH. In addition, in a dual-species or triple species environment harbouring *H. influenzae*, the potential release of ammonia to the extracellular environment could provide temporary neutralisation of organic acids produced as a result of *S. pneumoniae* fermentation.

Our results reveal the importance of arginine in the polymicrobial environment and strongly suggest the presence of a modified arginine deiminase pathway in *H. influenzae*, necessitating future investigations for a candidate gene with orthologous function to the arginine deiminase.

Arginine and ornithine, a product of arginine catabolism through the arginine deiminase pathway, have been increasingly shown to play an important role in other bacterial inter-species interactions. In the context of oral bacteria, co-aggregation of *Actinomyces naeslundii* has been shown to enable *Streptococcus gordonii* to survive in a arginine restricted environment [29]. Ornithine has been shown to function as a signalling molecule, and to drive the formation of a polymicrobial biofilm of *S. gordonii* and *Fusobacterium nucleatum* [30]. Therefore, it is possible that arginine plays a broader role in bacterial communication and in driving transition from planktonic to biofilm state for the otitis media polymicrobial biofilm. Our results, particularly with strain 86-028NP, showed a less pronounced growth inhibition in polymicrobial environments in a biofilm state compared to the planktonic state (Fig 3), suggesting that arginine may have a distinct role in the polymicrobial biofilm.

Overall, our analysis of the triple-species polymicrobial environment of *H. influenzae*, *S. pneumoniae* and *M. catarrhalis* has shown time-dependent competitive interactions between these bacterial species, and a strong requirement for arginine in permitting *H. influenzae* survival in a dual-species environment with *S. pneumoniae*, and in a triple-species environment. Arginine was shown to drive ATP production in *H. influenzae*, and increased ATP production was seen in dual-species and triple-species environments. These results highlight the critical role of arginine in the polymicrobial community and strongly suggest the presence of an arginine deiminase pathway in *H. influenzae*, which should be investigated in future studies.

## Supporting information

**S1 Fig. *artM* expression is increased in *H. influenzae* during co-culture with *S. pneumoniae*, and during triple-species culture.** Differential gene expression of *artM* in *H. influenzae* 86-028NP (86), *H. influenzae* in co-culture with *S. pneumoniae* (86+11), *H. influenzae* in co-culture with *M. catarrhalis* (86+QC), and *H. influenzae* in triple-species culture (Triple). Gene expression in presented as % gene expression relative to *artM* expression in 86-028NP in mono-culture. The data represents the average of 2 independent experiments, each performed in triplicate (*, $p < 0.1$).
(DOCX)

**S2 Fig. Arginine supplementation results in biphasic growth of *S. pneumoniae*, but does not impact *H. influenzae* or *M. catarrhalis* growth.** Planktonic growth of mono-, dual- and triple-species cultures of *H. influenzae* 86-028NP, *S. pneumoniae* 11 and *M. catarrhalis* QC at pH 8 over 18h, with (grey) and without (black) supplementation of exogenous arginine at a concentration of 4g/L. Growth is shown for *H. influenzae* 86-028NP in mono-culture (A), *S. pneumoniae* 11 in mono-culture (B), *M. catarrhalis* QC in mono-culture (C), dual-species culture of *H. influenzae* 86-028NP and *S. pneumoniae* 11 (D), dual-species culture of *H. influenzae* 86-028NP and *M. catarrhalis* QC (E), dual-species culture of *S. pneumoniae* 11 and *M. catarrhalis* QC (F), and triple-species culture (G).
(DOCX)

**S3 Fig. Arginine supplementation does not affect intracellular pH of *H. influenzae*.** Intracellular pH of *H. influenzae* following 30min incubation in the presence of 0, 2 or 4g/L of supplemented arginine.
(DOCX)

**S1 Table. Genes up-regulated in *H. influenzae* 86-028NP following 2 h growth in co-culture with *S. pneumoniae* 11.**
(DOCX)

**S2 Table. Genes up-regulated in *H. influenzae* 86-028NP following 2 h growth in triple species co-culture with *S. pneumoniae* 11 *and* *M. catarrhalis* QC.**
(DOCX)

**S3 Table. Genes down-regulated in *H. influenzae* 86-028NP following 2 h co-culture with *S. pneumoniae* 11.**
(DOCX)

**S4 Table. Genes down-regulated in *H. influenzae* 86 following 2 h growth in triple-species co-culture with *S. pneumoniae* 11 and *M. catarrhalis* QC.**
(DOCX)

## Acknowledgments

We acknowledge the contribution of Olivia Pei Ying Oh to assistance with initial growth experiments. We acknowledge Peter Sideris from SA Pathology for providing the reference strain of *Moraxella* sp.

## Author Contributions

**Conceptualization:** Alexandra Tikhomirova, Peter S. Zilm, Stephen P. Kidd.

**Data curation:** Alexandra Tikhomirova.

**Formal analysis:** Alexandra Tikhomirova, Claudia Trappetti, James C. Paton, Stephen P. Kidd.

**Funding acquisition:** Alexandra Tikhomirova, James C. Paton, Stephen P. Kidd.

**Investigation:** James C. Paton.

**Methodology:** Alexandra Tikhomirova, Claudia Trappetti, Stephen P. Kidd.

**Supervision:** Peter S. Zilm.

**Writing – original draft:** Alexandra Tikhomirova, Stephen P. Kidd.

**Writing – review & editing:** Alexandra Tikhomirova, Peter S. Zilm, Claudia Trappetti, James C. Paton, Stephen P. Kidd.

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
