## [Decision Letter · Decision Letter 0]

29 Jun 2022

PONE-D-22-10862The central role of arginine in Haemophilus influenzae survival in a polymicrobial environment with Streptococcus pneumoniae and Moraxella catarrhalis.PLOS ONE

Dear Dr. Kidd,

Thank you for submitting your manuscript to PLOS ONE and our apologies for the delay in the review process. Your manuscript was reviewed by two expert reviewers whose comments are included at the bottom of this message. Based on their recommendation, we feel that the manuscript has merit but does not yet fully meet PLOS ONE’s publication criteria as it currently stands. Therefore, we invite you to submit a revised version of the manuscript with minor revisions that address the points raised by the reviewers.

We look forward to receiving your revised manuscript.

Kind regards,

Hendrik W. van Veen, PhD

Academic Editor

PLOS ONE

Journal Requirements:

“This work was supported by the Garnett Passe and Rodney Williams Memorial Foundation Research Training Fellowship to A.T., and a National Health and Medical Research Council (NHMRC) Investigator Grant 1174876 to J.C.P.”

“This work was supported by the Garnett Passe and Rodney Williams Memorial Foundation Research Training Fellowship to A.T., and a National Health and Medical Research Council (NHMRC) Investigator Grant 1174876 to J.C.P”

 “This work was supported by the Garnett Passe and Rodney Williams Memorial Foundation Research Training Fellowship to A.T., and a National Health and Medical Research Council (NHMRC) Investigator Grant 1174876 to J.C.P.”

5. You should list all authors and all affiliations as per our author instructions and clearly indicate the corresponding author.

Reviewers' comments:

Reviewer's Responses to Questions

**Comments to the Author**

1. Is the manuscript technically sound, and do the data support the conclusions?

Reviewer #1: Yes

Reviewer #2: Yes

2. Has the statistical analysis been performed appropriately and rigorously? 

Reviewer #1: Yes

Reviewer #2: Yes

3. Have the authors made all data underlying the findings in their manuscript fully available?

Reviewer #1: Yes

Reviewer #2: Yes

4. Is the manuscript presented in an intelligible fashion and written in standard English?

Reviewer #1: Yes

Reviewer #2: Yes

5. Review Comments to the Author

Reviewer #1: In this manuscript, Tikhomirova and colleagues have studied in vitro competitive interactions among Haemophilus influenzae, Streptococcus pneumoniae and Moraxella catarrhalis, three pathogens that co exist in the human upper respiratory tract and also cause otitis media. They show decreased viability of H. influenzae in co-culture with S. pneumoniae and in triple-species cultures. They demonstrate a key role for arginine in mediating these interactions and further show that arginine stimulates production of ATP by H. influenzae in competitive cultures.

Overall the work is straightforward and well done and the data support the conclusions. A limitation of the work is that a limited numbers strains was studied. Individual strains of H. influenzae show substantial variability in growth characteristics and growth requirements between strains. The present work lays a good foundation to assess a broader range of clinical isolates of all three species as different results are likely to be seen. Just the two strains of H. influenzae yielded different results in the present study. The work also establishes observations and methods that will guide work under conditions that better simulate those in the human respiratory tract.

Reviewer #2: The authors present an interesting and well-conducted study describing the role of arginine for survival of Haemophilus influenzae in polymicrobial co-existence with S. pneumoniae and M. catarrhalis, relevant co-habitants of the upper respiratory tract and the inner ear, of importance for the development of otitis media.

A major comment:

In none of the clinical studies I have been involved in focused on bacterial colonization and co-occurrence in the upper respiratory tract I have ever observed a negative correlation between S. pneumoniae and H. influenzae (as opposed to the combination S. pneumoniae and S. aureus), in contrast we have found quite some evidence for synergy. In quite some other clinical studies this observation is also found. Why do the authors focus on the competition and observe reduction in viability when cultures of the different species are combined, while there is hardly or no evidence from human colonization studies for competition rather for inter-dependence? How do the authors explain this important difference between their in vitro results and the in vivo results described by others?

Minor comments:

-The sentence ‘conditions we previously identified resulted in reduced species competition’ (163-164) is not clear (despite the reference) the authors should elaborate a bit more to explain which conditions they refer to.

-It is not entirely clear why two unencapsulated H. influenzae strains were used. It is important to motivate this choice.

6. PLOS authors have the option to publish the peer review history of their article (what does this mean?). If published, this will include your full peer review and any attached files.

Reviewer #1: No

Reviewer #2: **Yes: **Marien I. de Jonge

---

## [Author Response · Author response to Decision Letter 0]

6 Jul 2022

Tikhomirova et al Rebuttal 

PONE-D-22-10862

The central role of arginine in Haemophilus influenzae survival in a polymicrobial environment with Streptococcus pneumoniae and Moraxella catarrhalis.

PLOS ONE

Reviewer #1

In this manuscript, Tikhomirova and colleagues have studied in vitro competitive interactions among Haemophilus influenzae, Streptococcus pneumoniae and Moraxella catarrhalis, three pathogens that co-exist in the human upper respiratory tract and also cause otitis media. They show decreased viability of H. influenzae in co-culture with S. pneumoniae and in triple-species cultures. They demonstrate a key role for arginine in mediating these interactions and further show that arginine stimulates production of ATP by H. influenzae in competitive cultures.

Overall the work is straightforward and well done and the data support the conclusions. A limitation of the work is that a limited numbers strains was studied. Individual strains of H. influenzae show substantial variability in growth characteristics and growth requirements between strains. The present work lays a good foundation to assess a broader range of clinical isolates of all three species as different results are likely to be seen. Just the two strains of H. influenzae yielded different results in the present study. The work also establishes observations and methods that will guide work under conditions that better simulate those in the human respiratory tract.

Response;

The point the Reviewer is making is very valuable. This study was performed as a focussed work aiming to establish an in-depth molecular characterisation of the interactions between S. pneumoniae, H. influenzae and M. catarrhalis, and it was unfortunately not within the scope of this work to include a deeper characterisation of a larger number of strains. However, the interesting results presented provide an alluring basis for the analysis of strain variation on the outcomes of the interactions between these three species, which we are eager to pursue in the future. 

Reviewer #2: The authors present an interesting and well-conducted study describing the role of arginine for survival of Haemophilus influenzae in polymicrobial co-existence with S. pneumoniae and M. catarrhalis, relevant co-habitants of the upper respiratory tract and the inner ear, of importance for the development of otitis media.

A major comment:

In none of the clinical studies I have been involved in focused on bacterial colonization and co-occurrence in the upper respiratory tract I have ever observed a negative correlation between S. pneumoniae and H. influenzae (as opposed to the combination S. pneumoniae and S. aureus), in contrast we have found quite some evidence for synergy. In quite some other clinical studies this observation is also found. 

Why do the authors focus on the competition and observe reduction in viability when cultures of the different species are combined, while there is hardly or no evidence from human colonization studies for competition rather for inter-dependence? How do the authors explain this important difference between their in vitro results and the in vivo results described by others?

Response:

We thank the reviewer for this interesting and thoughtful comment. There certainly are numerous studies suggesting cooperation between these bacterial otopathogens. However, it is worth noting that there are indeed opposing studies. We discuss this in lines 43-57. This is not only from in vitro and animal studies (ref 14, ref 16) but some indication from clinical studies. This has now been added into the revised introduction – line 51-55. 

For example, in children with upper respiratory tract infections, an antagonistic relationship between S. pneumoniae and H. influenzae has been suggested (ref 15 - Littorin et al, 2021). Ref 14 -Pettigrew et al, 2008 also observed negative associations between S. pneumoniae and H. influenzae during respiratory tract infection. In addition, clinical studies have shown strain specificity to affect inter-species interactions (ref 16 - Dagan et al, 2013), as well as significant variation in inter-species associations between anatomical sites, where although a high dominance of all three species was observed in the nasopharynx, the ear displayed increased dominance and abundance of H. influenzae compared to S. pneumoniae or M. catarrhalis (Smith-Vaughan et al, 2013 – ref 5). Clearly, in a clinical setting if there is a competitive outcome (such as, as a consequence of hydrogen peroxide being generated by S. pneumoniae inhibiting growth of the other bacterial species, as has been suggested, (ref 19 Ikryannikova et al, 2017)) then there will only be isolation and identification of S. pneumoniae from that sample, seemingly as a mono-culture colonisation or infection. The other bacteria that have not been cultured, exactly because they were outcompeted. 

There would seem to be opportunities for either synergistic or competitive interactions. This field has been equivocal, the outcomes of the interplay between bacteria seems to be dependent on specific factors. Indeed, our previous, published studies (ref 9 and 10) demonstrated that depending on environmental conditions (pH, nutrient availability and the growth phase) there is synergistic or competitive interactions between S. pneumoniae and H. influenzae. 

These are the references we mention above and are now added within our manuscript:

Littorin, N, et al., Decreased prevalence of Moraxella catarrhalis in addition to Streptococcus pneumoniae in children with upper respiratory tract infection after introduction of conjugated pneumococcal vaccine: a retrospective cohort study. 2021. Clinical Microbiology and Infection, 27(4):630e1-e6.

Pettigrew, M, M, et al., Microbial interactions during upper respiratory tract infections. 2008. Emerging Infections diseases, 14(10):1584–1591

Dagan, R, et al., Mixed Pneumococcal–Nontypeable Haemophilus influenzae otitis media is a distinct clinical entity with unique epidemiologic characteristics and pneumococcal serotype distribution. 2013. The Journal of Infectious Diseases, 208(7): 1152–1160. 

Minor comments:

-The sentence ‘conditions we previously identified resulted in reduced species competition’ (163-164) is not clear (despite the reference) the authors should elaborate a bit more to explain which conditions they refer to.

Response:

Specifically, we reported (ref 9; 10) that there can be either positive interactions or a negative, competitive interactions as a consequence of the distinct growth parameters: pH, nutrient availability or the stage of growth phase. We have added in this text to make this clear (lines 163-167). 

-It is not entirely clear why two unencapsulated H. influenzae strains were used. It is important to motivate this choice.

Response:

As noted by the Reviewers, this is an original study that creates a platform for further in-depth research. As such both the model H. influenzae laboratory strain Rd KW20 was used in (as performed in our previously published analyses, ref 9 and 10), as well as the genetically distinct, nontypeable, and clinical OM isolate 86-028NP. Nontypeable H. influenzae (NTHi) do not have a polysaccharide capsule and are a leading cause of OM. Furthermore, NTHi have specifically been associated with polymicrobial OM (Barkai, 2009), hence were highly relevant to the assessment of bacterial interactions in an OM context.

Barkai G, et al., Potential contribution by nontypeable Haemophilus influenzae in protracted and recurrent acutre otitis media. 2009. The Pediatric Infectious Disease Journal, 28(6): 466-471.

---

## [Editor Report · Decision Letter 1]

11 Jul 2022

The central role of arginine in Haemophilus influenzae survival in a polymicrobial environment with Streptococcus pneumoniae and Moraxella catarrhalis.

PONE-D-22-10862R1

Dear Prof. Kidd,

We’re pleased to inform you that your manuscript has been judged scientifically suitable for publication and will be formally accepted for publication once it meets all outstanding technical requirements.

Kind regards,

Hendrik W. van Veen, PhD

Academic Editor

PLOS ONE

---

## [Editor Report · Acceptance letter]

14 Jul 2022

PONE-D-22-10862R1 

The central role of arginine in *Haemophilus influenzae* survival in a polymicrobial environment with *Streptococcus pneumoniae* and *Moraxella catarrhalis.*

Dear Dr. Kidd:

I'm pleased to inform you that your manuscript has been deemed suitable for publication in PLOS ONE. Congratulations! Your manuscript is now with our production department. 

Kind regards, 

on behalf of

Prof. Hendrik W. van Veen 

Academic Editor

PLOS ONE